# The Effect of the Sodium—Glucose Cotransporter Inhibitor on Cognition and Metabolic Parameters in a Rat Model of Sporadic Alzheimer’s Disease

**DOI:** 10.3390/biomedicines11041025

**Published:** 2023-03-27

**Authors:** Jelena Osmanović Barilar, Ana Babić Perhoč, Ana Knezović, Jan Homolak, Davor Virag, Melita Šalković-Petrišić

**Affiliations:** Department of Pharmacology and Croatian, Institute for Brain Research, School of Medicine, University of Zagreb, 10000 Zagreb, Croatia

**Keywords:** sodium–glucose cotransporter inhibitor, streptozotocin, Alzheimer’s disease, glucagon-like peptide 1

## Abstract

Type 2 diabetes mellitus increases the risk of sporadic Alzheimer’s disease (sAD), and antidiabetic drugs, including the sodium–glucose cotransporter inhibitors (SGLTI), are being studied as possible sAD therapy. We have explored whether the SGLTI phloridzin may influence metabolic and cognitive parameters in a rat model of sAD. Adult male Wistar rats were randomized to a control (CTR), an sAD-model group induced by intracerebroventricular streptozotocin (STZ-icv; 3 mg/kg), a CTR+SGLTI, or an STZ-icv+SGLTI group. Two-month-long oral (gavage) SGLTI treatment (10 mg/kg) was initiated 1 month after STZ-icv and cognitive performance tested prior to sacrifice. SGLTI treatment significantly decreased plasma glucose levels only in the CTR group and failed to correct STZ-icv-induced cognitive deficit. In both the CTR and STZ-icv groups, SGLTI treatment diminished weight gain, decreased amyloid beta (Aβ) 1-42 in duodenum, and decreased the plasma levels of total glucagon-like peptide 1 (GLP-1), while the levels of active GLP-1, as well as both total and active glucose-dependent insulinotropic polypeptide, remained unchanged, compared to their respective controls. The increment in GLP-1 levels in the cerebrospinal fluid and its effect on Aβ 1-42 in duodenum could be one of the molecular mechanisms by which SGLTIs indirectly induce pleiotropic beneficial effects.

## 1. Introduction

Brain metabolism transforms with normal aging, and transient, dynamic metabolic insufficiency may underlie critical progression from aging into Alzheimer’s disease (AD). With age, the disbalance of brain metabolic needs and vascular-related substrate supply progresses, leading to age-related brain deterioration. This dynamic metabolic insufficiency can occur when there is intermittent supply–demand mismatch, as in the case of diabetes mellitus [1]. Nowadays, we know that diabetes mellitus increases the risk for developing sporadic Alzheimer’s disease (sAD) [2]. Therefore, drugs used in the treatment of diabetes are increasingly being studied as possible sAD therapies [3]. Determining whether diabetes medications reduce the risk of dementia is very complex. There are several approaches to treating insulin resistance in diabetes mellitus type 2 (T2DM). In addition to recommendations related to lifestyle changes, diet, and physical activity, treatment for T2DM includes and combines several classes of drugs, including sodium–glucose co-transporter 2 (SGLT2) inhibitors [4]. The potential effectiveness of only a number of these drugs in relation to dementia has been tested in observational studies (as monotherapies or as part of combination therapies), and even fewer have been tested in clinical trials in patients with AD [5]. The aforementioned drug class of SGLT2 inhibitors, which lower blood glucose levels by increasing the renal excretion of glucose into urine, were recently expanded with the approval of a dual inhibitor, which, in addition to blocking SGLT2, also inhibits a similar SGLT1 co-transporter. Among other tissues, SGLT1 is also found in the intestines, and its inhibition reduces glucose absorption in the small intestine after meals [6]. Some experimental preclinical studies have shown that the use of SGLT2 inhibitors can improve cognitive impairment caused by diabetes. There are only a few clinical studies investigating this effect which showed highly inconsistent results, and some of them showed no positive effect of SGLT2 inhibitors on cognitive functions [7,8]. Unfortunately, although SGLT2 is present in the brain, the effects of dual SGLT inhibition (dual SGLTIs inhibit both SGLT1 and 2) on cognitive deficit remain partly unknown. There is perhaps a greater chance for a better effect of dual inhibitors in the treatment of dementia, since there are more SGLT1 co-transporters in the brain than SGLT2 [9]. Several studies have confirmed the beneficial effect of phloridzin (a dual SGLT inhibitor) on cognitive deficits in different rat models of AD [10,11]. The effect of phloridzin was tested in rats treated with intracerebroventricular streptozotocin (STZ-icv) [10], which is used as a representative model for sAD associated with insulin resistance. STZ-icv, at low, non-diabetogenic doses, impairs cerebral glucose and energy metabolism, induces insulin resistance, causes neuroinflammation, and increases oxidative stress, all together inducing biochemical changes similar to those found in the brain of sAD patients [12,13,14,15,16,17]. At the histopathological level, STZ-icv-treated rats exhibit neuronal loss, amyloid plaques, and tau hyperphosphorylation in the hippocampus and cortex [18,19]. Additionally, a decrement in GLP-1 activity was observed in plasma and cerebrospinal fluid (CSF) samples of the STZ-icv rat model [20].

Hence, a nonselective SGLTI may influence metabolic and cognitive parameters in the STZ-icv rat model of sAD. The main goal of this study was to investigate the effect of dual SGLT1 and SGLT2 inhibition on cognition in the STZ-icv rat model of sAD. Additionally, given the efficacy of GLP-1 agonists and DPP-4 inhibitors in animal models and previous results indicating disrupted GLP-1 homeostasis in the STZ-icv model, it was important to determine whether the possible pleiotropic effects of dual SGLT inhibition depend on changes in GLP-1 signalling.

## 2. Materials and Methods

### 2.1. Animals and Experimental Design

Adult male Wistar Hann rats (University of Zagreb, School of Medicine (UZMS), Department of Pharmacology) were individually housed in cages with standard wood-chip bedding and standardized food pellets and water ad libitum. Animals were housed in a licensed animal facility at the Department of Pharmacology, UZMS (HR-POK-007), in conditions of standard temperature (22–25 °C) and humidity (40–60%), with a 12 h light/12 h dark cycle. At the age of 3 months, rats were randomly assigned to either the control group (CTR; *n* = 20) or the streptozotocin group (STZ; *n* = 20) and underwent STZ-icv/buffer-icv treatment (see below). One month after the treatment, daily oral gavage was initiated with 10 mg/kg phloridzin [21], which lasted for 2 months. Cognitive assessment was performed by Morris water maze (MWM) test before, and 1 month and 2 months after the start of phloridzin treatment. The passive avoidance test (PAT) was only conducted at the end of phloridzin therapy, as this test is not suited for repeated use. Intraperitoneal glucose tolerance test (ipGTT) was performed both prior to and following the completion of phloridzin treatment. At sacrification, blood and CSF samples were withdrawn from all subjects. Animals were weighed on a regular basis during SGLTI treatment to keep track of weight changes. The experimental design is pictured in Figure 1.

### 2.2. Materials

ELISA Kits for rat/mouse insulin, active GLP-1, total GLP-1, and total GIP were purchased from Merck Millipore (Billerica, MA, USA). The ELISA kit for active GIP from Immuno-Biological Laboratories, Inc. (Minneapolis, MN, USA). The GOD-PAP glucose measuring kit was acquired from Dijagnostika (Sisak, Croatia). Accu-Check Performa glucometer and test strips (Roche, Switzerland) from Medical Intertrade d.o.o. (Sveta Nedelja, Croatia). Ketamidor 10% (ketamine hydrochloride, Richter Pharma, Wels, Austria), Xylazine 2% (xylazine chloride, Alfasan International B.V., Woerden, The Netherlands), Heparin Belupo (heparine sodium 25 000 IU/5 mL, Belupo d.d., Koprivnica, Croatia) from Medical Intertrade d.o.o. (Sveta Nedelja, Croatia). The SGLT inhibitor phloridzin, DPP-4 inhibitor, and streptozotocin were acquired from Sigma Aldrich-Merck KGaA (Darmstadt, Germany). Infinite F200 PRO multimodal microplate reader (Tecan, Männedorf, Switzerland) was used for colorimetric/fluorometric readings.

### 2.3. Drug Treatments

Streptozotocin (STZ) treatment. At the age of 3 months, rats were subjected to general anaesthesia (ketamine 70 mg/kg/xylazine 7 mg/kg ip), and administered STZ-icv bilaterally into the lateral ventricles (2 µL/ventricle) using the following coordinates: −1.5 mm posterior; ±1.5 mm lateral; +4 mm ventral from the pia mater relative to bregma, in a total dose of 3 mg/kg (dissolved in 0.05 M citrate buffer, pH 4.5, split in two doses on day 1 and day 3, with control animals receiving citric buffer only in the same manner), according to a well-established procedure repeatedly used in our previous research [22].

Phloridzin (SGLTI) treatment. Phloridzin was dissolved ex tempore in tap water in a concentration of 10 mg/kg and administered daily for a total of 2 months, in a total volume of 1 mL, to conscious rats via oral gavage, using flexible plastic feeding tubes, to CTR+SGLTI and STZ+SGLTI groups. CTR and STZ groups drank tap water ad libitum. SGLTI treatment was initiated 4 weeks after STZ-icv administration. Considering that a significant reduction in hyperglycaemia and improvement of disturbed lipid metabolism, associated with minor side effects, were obtained following the 10 mg/kg oral dose of phloridzin [21], we have chosen this dose (10 mg/kg) to explore its effect on cognition [21].

### 2.4. Cognitive Testing

Morris water maze (MWM) test. MWM was performed at 3 time-points, before and after the 30 and 60 days treatment with SGLTI. The test [23] lasted for a total of 6 days; 5 learning and memory trial days with 4 trials per day (each from a different starting point of the virtually divided pool quadrants; southwest (SW), south (S), east (E), and northeast (NE), separated by 30 min of rest periods), and a probe trial on day 6. Using a round (180 cm diameter) pool filled with water at a temperature of 25 ± 1 °C, 60 cm deep, on days 1 to 5 animals were trained to escape water by finding a hidden, 15 cm diameter wide, glass platform submerged 2 cm below water surface. Each training day, the platform was placed in the middle of the northwestern (NW) quadrant and, once the platform was found by the swimming animal, each rat was allowed to remain on the platform for 15 s to memorize its location. The time needed to find the platform after the start of the trial (i.e., escape latency) was recorded in every training trial to assess the aptitude of learning memory. In the probe trial on day 6, the animals’ memory retention was tested by starting all trials from the southeastern (SE) quadrant with the platform removed from the pool. Time spent in search of the platform in the NW quadrant was recorded. In addition, non-target zone entries (entries to quadrants other than NW; i.e., “number of errors”) were recorded for all trials. Animals with memory impairment were expected to have diminished memory of the platform’s location in the NW quadrant, and consequently, to spend less time in search of the platform within the quadrant, with more erroneous entries to non-target quadrants. Data were acquired by Basler AG camera and tracked and analysed by EthoVision XT software (Noldus Information Technology, Wageningen, The Netherlands).

Passive Avoidance test (PAT). The test is performed using a step-through-type passive avoidance apparatus (Ugo Basile, Comerio, Italy). PA behaviour was analysed by exploiting the rodents’ innate preference for dark environments [24,25]. The test is a fear-motivated avoidance task in which animals refrain from stepping through a sliding door to an apparently safer, dark but previously punishment-related compartment. The latency to avoid stepping into the aversive compartment is used as an index of the ability to avoid and allows evaluation of memory. The test was performed during three days after treatment with SGLTI. On day 1, the animals were left to familiarize themselves with the environment (without foot shock, i.e., pre-shock latency), which was followed by a training day on day 2, when a foot shock (0.3–0.5 mA, depending on the animal weight, duration 2 s) was delivered, and, lastly, on day 3 (no foot shock, i.e., post-shock latency) the time required to enter the dark compartment was recorded with a cut-off time of 5 min. Animals without memory impairment (controls) were expected to remember receiving the shock upon crossing into the dark compartment and, consequently, to remain in the light compartment for a longer period of time.

### 2.5. Intraperitoneal Glucose Tolerance Test (ipGTT)

To assess the integrity of the animals’ glucose metabolism function, ipGTT was performed before and after SGLTI treatment. The procedure [26,27] was conducted by obtaining free-flowing blood from the animal’s tail tip with sterile scissors before and after a 2 g/kg body weight glucose load (20% glucose solution in saline) administered intraperitoneally. Blood glucose levels were measured using a glucometer and disposable test strips (F. Hoffmann-La Roche AG, Basel, Switzerland) before the load and 15, 30, 45, 60, and 120 min after.

### 2.6. Biochemical Analyses

Total GLP-1. A commercial ELISA kit with declared sensitivity of 1.5 pM was used to measure plasma and CSF levels of total GLP-1 (GLP-1 7-36 and 9-36. The procedure was performed in accordance with the manufacturer’s protocol. Absorbance was measured at 450 nm with reference measurement at 590 nm using a microplate reader. The levels of total GLP-1 are expressed in pmol/L (pM).

Active GLP-1. Active GLP-1 (GLP-1 7-36 and 7-37) levels in DPP-IV inhibitor pre-treated plasma samples were assessed using a commercial ELISA kit with a declared sensitivity limit of 2 pM. The procedure was carried out by adhering to the manufacturer’s protocol. Results were obtained using a fluorescence plate reader at an excitation/emission wavelength of 355/460 nm. The levels of active GLP-1 are expressed in pmol/L (pM). The GLP-1 active ELISA kit quantifies active forms of GLP-1 (7-36 amide and 7-37) and does not detect other forms, while the GLP-1 total ELISA kit is intended for quantification of both active (7-36) and non-active (9-36; degraded by dipeptidyl peptidase IV) forms.

Total GIP. Levels of both GIP (1-42) and GIP (3-42) were measured in plasma and CSF samples using a commercial ELISA kit for total GIP, with a declared sensitivity limit of 8.2 pg/mL, and according to the attached procedure protocol. The enzyme activity was measured spectrophotometrically by the increased absorbency at 450 nm, corrected from the absorbency at 590 nm, after acidification of formed products. Results are expressed as pg/mL.

Active GIP. The active moiety of GIP (1-42) was measured in plasma samples pre-treated with a DPP-IV inhibitor immediately after sampling, using a commercial high-sensitivity ELISA kit with 0.06 pmol/L sensitivity. The analysis was carried out according to the manufacturer’s instruction and the results were measured at 450 nm, with reference between 600 and 650 nm. Data are expressed as pmol/L (pM).

Glucose. Plasma and CSF glucose levels were measured using the colorimetric GOD-PAP method, in which glucose is oxidized by glucose oxidase to gluconic acid and hydrogen peroxide in conjunction with peroxidase reacts with chloro-4-phenol and 4-amino-antipyrine to form a red quinoneimine, the absorbance of which is measured at 500 nm (Trinder method). Results are expressed as mmol/L.

Tissue preparation. Fresh frozen hippocampi (HPC) were thawed and ultrasonically homogenized with lysis buffer (10 mM HEPES, 1 mM EDTA, 100 mM KCl, 1% Triton X-100, pH 7.5) with protease phosphatase inhibitor cocktails (1:10; Sigma-Aldrich, Burlington, MA, USA and PhosSTOP, F. Hoffmann-La Roche AG, Basel, Switzerland). Proximal duodenal segments (1 cm distal to pylorus) were excised, cleaned of intraluminal contents with ice-cold PBS, and homogenized in lysis buffer (150 mM NaCl, 50 mM Tris-HCl pH 7.4, 1 mM EDTA, 1% Triton X-100, 1% sodium deoxycholate, 0.1% SDS, 1 mM PMSF, protease (Sigma-Aldrich, Burlington, MA, USA) and phosphatase (PhosSTOP, F. Hoffmann-La Roche AG, Basel, Switzerland) inhibitor cocktails; pH 7.4) with an ultrasonic homogenizer (Microson Ultrasonic Cell 167 Disruptor XL, Misonix, Farmingdale, New York, NY, USA). Homogenates were centrifuged at 4 °C for 10 min at a relative centrifugal force of 12 879 g. Protein concentration used for correction was measured with the Bradford reagent (Sigma-Aldrich, St. Louis, MO, USA) following the manufacturer’s instructions.

Soluble Aβ1-42. Levels of soluble Aβ1-42 in hippocampal and duodenal homogenates were measured using a commercial ELISA kit for human and rat amyloid beta peptide 1-42, with a measurement range of 12.35 pg/Ml–1000 pg/mL and sensitivity <4.96 pg/mL, in accordance with the manufacturer’s protocol. Absorbance was measured at 450 nm using a microplate reader. The obtained concentration was expressed in pg/mL per total protein.

Catalase activity. Catalase activity was estimated from H_2_O_2_ dissociation rate using modified Hadwan’s procedure [28], described in detail in [29]. Briefly, tissue homogenates (15 μL HPC; 7 μL duodenum) were incubated with 150 μL of the Co(NO_3_)_2_ solution (0.1 g of Co(NO_3_)_2_ × 6 H_2_O in 5 mL ddH_2_O mixed with 0.05 g (NaPO_3_)_6_ dissolved in 5 mL ddH_2_O; added to 90 mL of NaHCO_3_ solution (8.1 g in 90 mL ddH_2_O)) followed by 50 μL 10 mM H_2_O_2_ in PBS to obtain baseline values. Next, homogenates were incubated with 10 mM H_2_O_2_ for 60 (duodenum) or 120 s (HPC) and the reaction was stopped with Co(NO_3_)_2_ solution. Catalase activity was estimated indirectly by quantification of Co oxidation in the presence of bicarbonate which reflects the baseline and final concentration of H_2_O_2_. Co oxidation was assessed by measuring the absorbance of [Co(CO_3_)_3_]Co at 450 nm with the Infinite F200 PRO multimodal microplate reader (Tecan, Switzerland). H_2_O_2_ concentration was estimated from a linear model obtained by repeating the same procedure with graded nominal substrate concentrations. Optimal homogenate volumes and reaction times were assessed for each tissue individually. Assay validation has been reported previously [30,31].

### 2.7. Ethics

All procedures involving animals and animal samples were performed in compliance with current institutional (UZSM), national (Animal Protection Act, NN 102/17, 32/19, 125/13, 14/14, 92/14), and international (Directive 2010/63/EU and Council Regulation (EC) No 1099/2009) guidelines on the use of experimental animals. The experiments were approved by the ethical committee of the UZSM and the national regulatory body, the Croatian Ministry of Agriculture (approval number: EP 186/2018; 380-59-10106-18-111/173).

### 2.8. Statistics

Data were expressed as mean ± SD and analysed using the student’s *t*-test for analysis of 2 groups and the non-parametric Kruskal–Wallis (KW) one-way analysis of variance for analysis of 4 groups, followed by Mann–Whitney U-test or Wilcoxon signed rank test. Longitudinal data were analysed by two-way repeated measures ANOVA (RM ANOVA) with Sidak, Bonferroni, or Tukey post-tests, with *p* < 0.05 considered as statistically significant. Catalase activity was analysed by modelling the difference between baseline and final H_2_O_2_ concentration as the dependent variable and group, protein concentration (loading control), and baseline concentration of H_2_O_2_ as independent predictors. Data were analysed using GraphPad Prism 8 statistical software (GraphPad Software, Inc., San Diego, CA, USA) and R (4.1.3; R Foundation for Statistical Computing, Vienna, Austria).

## 3. Results

### 3.1. Body Weight

Animals were weighed before initiating SGLTI treatment and twice weekly after the start of the treatment. Before initiating treatment, STZ animals showed statistically significant decreased weight compared to controls, one month after STZ treatment (−6.7% compared to CTR, *p* = 0.0114; Figure 2a). After initiating treatment, SGLTI-treated animals showed a trend of decreased weight gain compared to their respective controls drinking tap water only (Figure 2b). A disturbance in the otherwise steady weight gain in all groups was noted at the one-month treatment time-point, when the MWM test was performed for a week.

### 3.2. Cognitive Assessment

MWM was conducted at 3 time-points: before, one month into SGLTI treatment, and following the 2-month-long SGLTI treatment. The effect of STZ on the rats’ learning abilities was evident 1 month following STZ treatment, when CTR animals retained intact learning abilities during the 5-day training period of the MWM test compared to STZ-treated rats, as evidenced in the decreased number of erroneous entries to non-target quadrants on trial days 3, 4, and 5 (−46.7%, −44.5%, and −40.1%, respectively, vs. STZ, *p* < 0.05; Figure 3b). At both the second and third time-points, 30 and 60 days after starting the treatment, there was no benefit in learning abilities from SGLTI treatment in either the STZ or the CTR group, and both the STZ and the STZ+SGLTI groups showed weaker learning performance in the MWM, as evidenced by both the increased time to find the platform and the number of erroneous entries to non-target quadrants (Figure 3c–f). The passive avoidance test confirmed these results and showed there was no beneficial effect on fear-conditioned memory in either the CTR or STZ group (Figure 3g).

### 3.3. ipGTT

ipGTT was performed in all rats before SGLTI treatment and confirmed there was no disturbance in glucose tolerance due to STZ-icv treatment (Figure 4a). After the 2-month-long SGLTI treatment, glucose tolerance was unchanged in all tested groups (Figure 4b).

### 3.4. Biochemical Analyses

Plasma samples were analysed for both total and active GLP-1 and GIP levels, as well as glucose levels, while only glucose, total GLP-1, and total GIP levels were assayed in CSF samples due to the limited volume of sampled CSF and below-detectable concentrations of active forms of both GLP-1 and GIP in the CSF.

SGLTI treatment showed significant lowering of total GLP-1 levels in CTR+SGLTI rats, as compared to both CTR (−35.8%, *p* = 0.0279) and STZ (−39.8%, *p* = 0.00288) (Figure 5a). In the CSF, levels of total GLP-1 were increased by SGLTI treatment in both CTR and STZ rats compared to CTR alone (+73.1%, *p* = 0.0262; +58.7%, *p* = 0.0343; respectively; Figure 5b). SGLTI treatment had no effect in either CTR or STZ animals on the levels of total GIP in either plasma or CSF nor on the levels of the active forms of GLP-1 or GIP (Figure 5c–f). As predicted by its mechanism of action, plasma glucose levels were significantly altered by SGLTI treatment; SGLTI treatment lowered plasma glucose levels in both CTR+SGLTI (−50.7%, *p* = 0.00065) and STZ+SGLTI groups (−39.3%, *p* = 0.0892), as compared to CTR (Figure 5g). In the CSF, no change was measured in glucose levels upon SGLTI treatment (Figure 5h).

Hippocampal and duodenal samples were analysed for Aβ content and catalase activity. Catalase activity was not significantly altered in the hippocampus (Figure 6a); however, in the duodenum, STZ decreased catalase activity and phloridzin showed a propensity to increase it (Figure 6b). Similarly, no differences were detected in hippocampal Aβ levels (Figure 6c), while duodenal Aβ1-42 content was altered following phloridzin treatment (−59.6%, *p* = 0.0043, CTR+SGLTI vs. CTR; −73.9% STZ+SGLTI vs. STZ, *p* = 0.0556; Figure 6d).

## 4. Discussion

The present study was undertaken to evaluate the effect of chronic treatment with the natural occurring polyphenol phloridzin, a potent nonselective SGLTI, on STZ-icv-induced cognitive deficit and its effect on duodenal Aβ 1-42 and catalase homeostasis in a sporadic rat model of Alzheimer’s disease. The results additionally provide preliminary evidence of changes in incretins level due to treatment with a non-selective SGLTI but without any beneficial effect of SGLT inhibition on cognition.

### 4.1. Chronic Effect of Low Dose of Phloridzin/Phloretin on Cognition

The neuroprotective effects of SGLTI have been described in animal models of AD and in some clinical studies [8,10,11,32]. However, in the present study, the beneficial effects of phloridzin, a nonselective SGLTI and an important compound from which all the selective SGLT2 inhibitors are derived from, were not detected.

There are several reasons that might play a role in this lack of effect. One possible reason may include the pharmacokinetic properties of phloridzin. Studies in which phloridzin exerted beneficial effects on cognition used the intraperitoneal route of administration [32,33], whereas, in the present experiment, phloridzin was administered orally, which infers a different pharmacokinetic profile and drug metabolism. After oral administration, phloridzin is hydrolyzed by β-glucosidase (EC 3.2.1.21) and lactase-phloridzin hydrolase (EC 3.2.1.62) to phloretin and glucose [34,35] on the brush border of the small intestine (predominantly in the jejunum), also involved in the degradation of lactose [34,36]. Compared with oral administration, intravenously or intraperitoneally given phloridzin avoids the first pass through the liver and degradation in the small intestine, so in the blood the predominant active compound is phloridzin and, in a lesser amount, phloretin [37]. Indeed, analysis of plasma samples of rats fed with phloridzin showed only the phloretin metabolite, indicating that the majority of systemic effects seen with oral chronic administration of phloridzin is due to the metabolite phloretin [38]. Phloretin has a different biological effect in comparison to phloridzin: (i) phloretin blocks GLUT-1 and -2 transport and facilitates diffusion of urea and glycerol, and also inhibits membrane transport of chloride, bicarbonate, and lithium ions in mammalian erythrocytes [39], (ii) it has been shown to possess modest oestrogen activity [40], (iii) it can affect mitochondrial oxidative phosphorylation [41] and (iv) has a lower affinity to SGLT 1 and 2 in comparison to phloridzin. Normally, phloridzin does not affect glucose uptake in the brain, because SGLT2 (expressed on the blood–brain barrier (BBB) endothelial cells) is not involved in glucose uptake by the brain [42]. The major glucose transporter in the brain is GLUT-1, which is inhibited by phloretin administration and which could explain the lack of beneficial effect of orally given phloridzin in this study, as evidenced both in spatial memory and fear-conditioned memory [43].

Data from the literature show that studies involving the oral route of administration of phloridzin were carried out on different models; e.g., diabetes-induced cognitive deficit or models mimicking neuroinflammation (LPS-induced cognitive deficit) [44]. Diabetes was induced by a high streptozotocin (STZ) dose given intraperitoneally together with high-fat diet. Afterwards, the animals were treated orally with phloridzin (10 and 20 mg/kg) for four weeks. Memory functions were evaluated by passive avoidance test and novel object recognition test. Both doses of phloridzin significantly reversed diabetes-induced memory impairment [45]. On the other hand, the study in the STZ-icv rat model treated with 20, 50, and 100 mg of phloridzin intraperitoneally showed a dose-dependent response [10]. It has been also demonstrated that phloridzin given intraperitoneally (10, 20 mg/kg) to mice with neuroinflammation induced by icv application of lipopolysaccharide (LPS) also showed an attenuation of cognitive deficit that was evaluated by Morris water maze (MWM) and Y-maze tests [44].

Additionally, the potential impact of gender on brain function is now emphasized in the field of AD research. Phloridzin and phloretin, two natural compounds with oestrogenic activity, can potentially regulate pro-oxidative and inflammatory mechanisms, which could influence the sex-different effect of the drug in the STZ-icv model [46]. Most studies with phloridzin to date in the AD model were performed on male rats and we agree that it is important to determine whether reproducible results could be found in females. Limitations of the study also include the use of only one dose of phloridzin.

In conclusion, the beneficial effect of phloridzin seems to depend on some drug-related issues such as the route of administration and dose, as well as on the model used to induce cognitive deficit. In the STZ-icv rat model of Alzheimer’s disease in our experiments, the low dose of 10 mg/kg given by oral gavage did not exert an effect on cognition, but it cannot be excluded that different doses (20–100 mg/kg) [10] might have a positive effect on cognitive functions in this model.

### 4.2. Chronic Effect of Low Dose of Phloridzin/Phloretin on Glucose and GLP-1 Levels and Body Weight

SGLT2 inhibitors show many additional beneficial effects which contribute to their wider use [47]. One of these beneficial effects is a reduction in body weight due to decreased plasma glucose levels induced by inhibition of glucose reabsorption in the kidneys and absorption in the gut [47,48]. As expected, in our study, decreased glucose plasma levels were observed in plasma of control and STZ-icv animals, indicating that the peripheral effects of phloridzin were not affected by the STZ-icv treatment (Figure 5g). The body weight of all rats treated with SGLTI showed less weight gain compared to controls (Figure 2b), which is in line with other studies [47,49]. This decrement in weight gain is related to decreased plasma levels of glucose (increased calorie loss) but was smaller in size than expected, possibly due to the activation of compensatory increases in appetite/caloric intake [48]. The decrement in circulatory glucose concentration results in mobilization of lipid storage, leading to changes in fuel substrate use, favouring the consumption of lipids for energy production [48,50]. These changes induce increases in lipolysis and production of ketone bodies leading to a metabolic condition similar to prolonged fasting [51]. The results of the performed ipGTT were not affected by chronic low-dose SGLTI administration either in the control or STZ-icv-treated group, which is in line with a Japanese study in which ipGTT changes due to phloridzin administration were dose-dependent [52].

The SGLTI treatment decreased total GLP-1 in plasma of both groups, but no changes in the levels of active form were observed (Figure 5a–c). This decrease in secretion of total GLP-1 is probably, as it seems from our results, compensated by changes in degradation by dipeptidylpeptidase-4 (DPP4), thus keeping the level of active GLP-1 in balance [53]. Our results are in line with other studies and are expected given the reduced blood glucose level [54,55,56]. More specifically, GLP-1 action is dependent on glucose; therefore, GLP-1 will be secreted only when glucose levels are increased. Additionally, GLP-1 will decrease glucose levels only when concentration is above the feasting levels [57,58]. The relative contribution of peripherally versus centrally produced GLP-1 in influencing brain functions is still unclear [59,60]. It seems that SGLTI reduces glucose absorption in the intestines, decreases glucose and GLP-1 levels in plasma, but unexpectedly increases the GLP-1 level in the CSF of treated controls. GLP-1 released from L cells acts locally and can activate the GLP-1R located on vagal sensory neurons. These vagal neurons transduce the signal to the brainstem and hypothalamus neurons, helping the brain to regulate the whole-body metabolism [61,62]. Endogenous GLP-1 appears to cross the BBB by a simple diffusion and has no other means of leaving the brain other than reabsorption to the CSF [63]. In line with these findings, increased CSF levels of GLP-1 found here in the control group could be a compensatory reaction of the decreased plasma level which is not seen in the STZ-icv-treated rats (Figure 5b). As GLP-1 has neuroprotective effects [3], the increase in GLP-1 in the CSF could be one of the mechanisms by which SGLTIs induce beneficial effects in the brain. In our previous research, we observed different responses in GLP-1 and GIP levels in plasma and CSF in the STZ-icv rat model of AD after treatment with exenatide (Ex9) [64]. Based on the highly increased GIP levels in plasma found after acute central GIPR inhibition, and the unchanged plasma GLP-1 levels after central GLP-1R inhibition, we can speculate that the CNS is more dependent on the production of GIP, i.e., more sensitive to a central dysfunction in GIP than GLP-1 homeostasis. Additionally when GLP-1 receptor antagonist Ex9 was given icv to the STZ-icv rat model of AD, it caused perturbations of the gastrointestinal tract, indicating an importance of the homeostasis in the brain–gut GLP-1 axis [65]. However, the exact role of SGLT inhibition in the regulation of this axis still remains to be determined.

### 4.3. Chronic Effect of Low Dose of Phloridzin/Phloretin on Aβ 1-42 and Catalase Interplay in the Hippocampus and Duodenum

New data indicate that Aβ load in the periphery might promote its accumulation in the brain due to its retrograde transport to the CNS via the vagal nerve [66,67]. In the gastrointestinal tract, Aβ is involved in nutrient (e.g., cholesterol) absorption and its expression is promoted by proinflammatory stimuli [68]. Furthermore, Aβ peptides are involved in the regulation of central insulin and glucose metabolism [69]. In the duodenum, phloridzin decreased Aβ in controls and STZ-icv animals. There was a tendency for increment in Aβ 1-42 in the STZ-icv group but because of the big intragroup variability, the difference was not statistically significant (Figure 6d). This finding is in line with data from the literature on the effect of glucose on Aβ secretion from adipocytes [70]. The mechanism behind the reduction in Aβ 1-42 after SGLTI treatment could be associated to diminished glucose absorption in the duodenum, which in turn could lead to decreased Aβ production. As expected, the level of Aβ 1-42 was not changed 3 months after STZ-icv treatment in the hippocampus. This is in line with our previous experiments which demonstrated that 3 months after STZ-icv, pathological Aβ accumulation is seen mainly as cerebral angiopathy [19,71]**,** while at this time-point, intraneuronal Aβ 1-42 pathological accumulation could be seen only in the parietal cortex and progression to the hippocampal region and formation of extraneuronal plaque-like formations were found no earlier than 6 months after STZ-icv [18]. In addition, the treatment with SGLTI did not have the lowering effect on Aβ 1-42 in the HPC as in the duodenum (Figure 6c,d).

Catalase, an antioxidative enzyme, is co-localized within senile plaques and can protect cells from Aβ 1-42 toxicity [72]. Catalase is responsible for the conversion of hydrogen peroxide (H_2_O_2_) formed by degradation of reactive oxidative species (ROS) to water and oxygen. Diminished catalase activity is associated with oxidative stress, aging, and Alzheimer’s disease. Normal activity of catalase is essential for maintaining an optimal level of H_2_O_2_ in the cell, which is also essential for cellular signalling processes [73,74]. Catalase activity was decreased in the duodenum, which is associated with increased Aβ 1-42 in the STZ-icv group (Figure 6d). Aβ 1-42 binds catalase in nanomolar levels which points to a potential physiological role of Aβ 1-42 as its inhibitor [75]. Our findings point to the importance of the homeostatic interplay between catalase and Aβ 1-42 in the duodenum of the STZ-icv-induced AD model. It cannot be ruled out that 3 months after STZ-icv administration, increased duodenal Aβ 1-42 could travel through the *nervus vagus* to the brain, thus increasing the probability of plaque formation in the brain in later time-points (6 months) after STZ-icv treatment [67]. Aβ 1-42 inhibition of catalase can increase cell concentrations of H_2_O_2_, leading to cellular damage causing cell death [73], which may support our recent findings of damaged gastrointestinal [76] and mucus barrier [77] and increased oxidative stress in the STZ-icv duodenum [65]. The presented results offer preliminary evidence of the involvement of SGLT in the regulation of Aβ 1-42 expression and catalase activity in the gastrointestinal tract and indicate that redox dyshomeostasis in the STZ-icv model of AD might be abolished with SGLTIs.

## 5. Conclusions

Our findings suggest that long-term treatment with low-dose nonselective SGLTI by phloridzin has no effect on cognition but has the ability to change the levels of GLP-1 in plasma and CSF, lower plasma glucose levels, and reduce body weight. The measured increment in GLP-1 levels in the CSF and its effect on Aβ 1-42 in the duodenum could be one of the molecular mechanisms by which SGLTIs indirectly induce pleiotropic beneficial effects in the brain. The decrement in plasma glucose and reduction in body weight contribute to the beneficial effects on the cardiovascular system seen in humans, but also on the brain. Further research is needed to elucidate the exact role of phloridzin dose and route of administration as well as the animal models used to test its effects for a more successful planning of non-clinical and clinical studies.

## Figures and Tables

**Figure 1 biomedicines-11-01025-f001:**
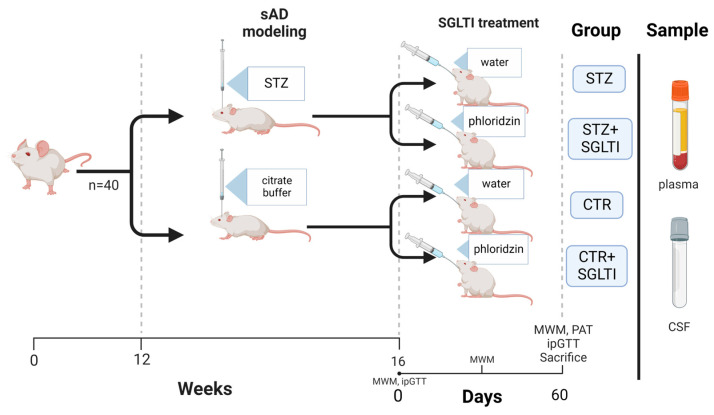
Experimental design. sAD—sporadic Alzheimer’s disease; STZ—streptozotocin group; SGLTI—sodium–glucose co-transporter inhibitor; MWM—Morris water maze test; PAT—passive avoidance test; CTR—control group; ipGTT—intraperitoneal glucose tolerance test; CSF—cerebrospinal fluid. Created with BioRender.com.

**Figure 2 biomedicines-11-01025-f002:**
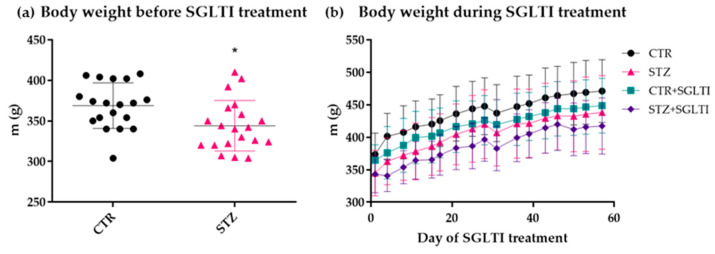
Body weight before initiation (**a**) and during (**b**) phloridzin treatment. Rats were weighed regularly (2× per week) during the 2-month-long phloridzin treatment. Data presented as mean ± SD with individual animal data and analysed by the student’s *t*-test (**a**) and presented as mean ± SD and analysed by 2-way repeated measures ANOVA with Tukey’s multiple comparisons test between groups (**b**). bw—body weight; SGLTI—sodium–glucose co-transporter inhibitor; CTR—control group; STZ—streptozotocin-treated group; CTR+SGLTI—control group treated with phloridzin; STZ+SGLTI—streptozotocin-treated group treated with phloridzin. * *p* < 0.05.

**Figure 3 biomedicines-11-01025-f003:**
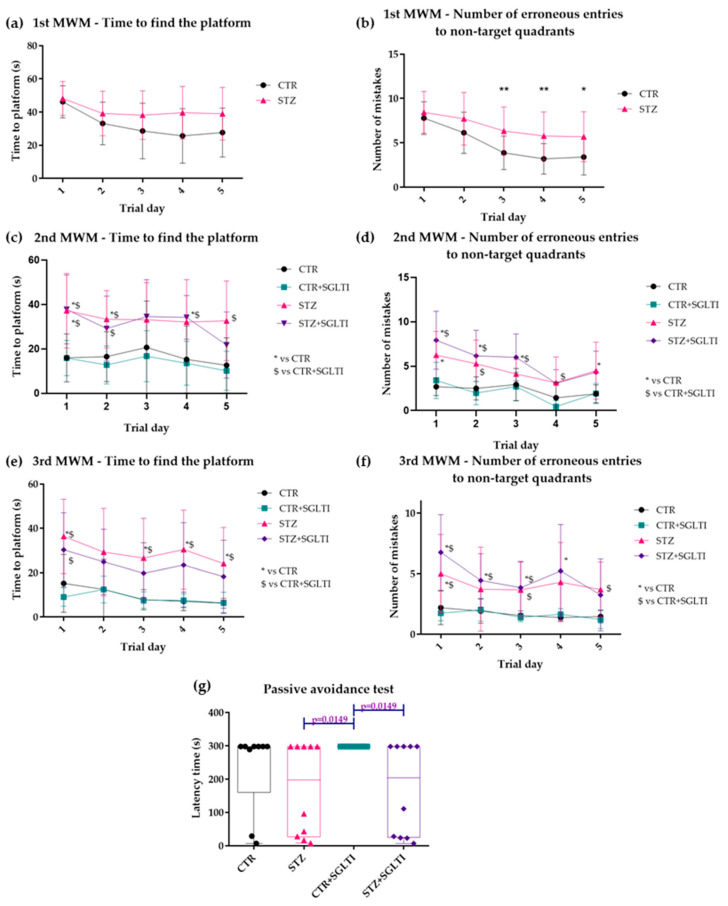
Cognitive assessment by Morris water maze (MWM) and Passive Avoidance (PAT) tests. MWM was conducted 1 month after intracerebroventricular administration of streptozotocin (STZ-icv) before the initiation of therapy with the sodium–glucose co-transporter inhibitor (SGLTI) phloridzin (**a**,**b**), 1 month into daily phloridzin therapy (**c**,**d**), and after 2 months of daily oral gavage with 10 mg/kg phloridzin solution (**e**,**f**). PAT was conducted only following 2 months of phloridzin (**g**). In the MWM test, both the time to find the platform during the 5 training days (**a**,**c**,**e**) and the number of erroneous entries to non-target quadrants (those not containing the hidden platform; i.e., the northwestern quadrant) (**b**,**d**,**f**) was recorded. In the PAT, the latency time to enter the previously punishment-related dark compartment of the apparatus on the 3rd day of the PAT was recorded (**g**). Data are presented as mean ± SD (**a**–**f**) or boxplots (mean with min and max showing individual data) (**g**). Data were analysed by 2-way repeated measures ANOVA with Bonferroni’s multiple comparisons for the 1st MWM (**a**,**b**) and Tukey’s multiple comparisons test between groups for the 2nd and 3rd MWM (**c**–**f**) and Kruskal–Wallis 1-way ANOVA with Wilcoxon signed rank tests for between-group comparisons (g). * *p* < 0.05 vs. CTR; ** *p* < 0.01 vs. CTR; ^$^
*p* < 0.05 vs. CTR+SGLTI. CTR—control group; STZ—streptozotocin-treated group; CTR+SGLTI—control group treated with phloridzin; STZ—streptozotocin-treated group treated with phloridzin.

**Figure 4 biomedicines-11-01025-f004:**
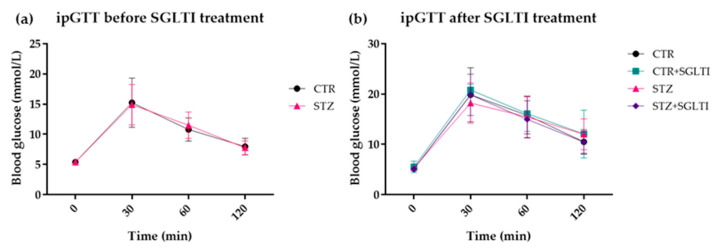
Results of the intraperitoneal glucose tolerance tests (ipGTT) performed before (**a**) and after phloridzin treatment. Data presented as mean ± SD and analysed by 2-way repeated measures ANOVA with Sidak (**a**) and Bonferroni’s (**b**) multiple comparisons test between groups. SGLTI—sodium–glucose co-transporter inhibitor; CTR—control group; STZ—streptozotocin-treated group; CTR+SGLTI—control group treated with phloridzin; STZ+SGLTI—streptozotocin-treated group treated with phloridzin.

**Figure 5 biomedicines-11-01025-f005:**
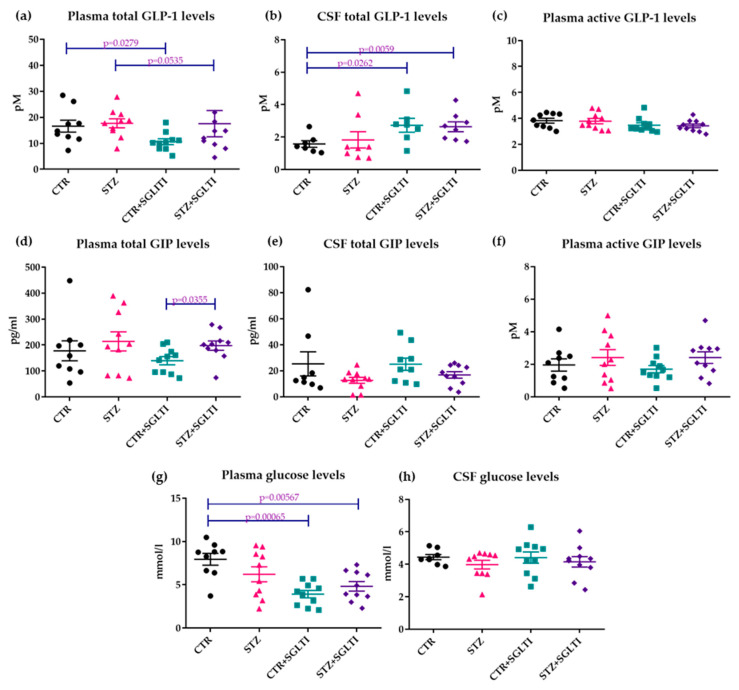
Biochemical analyses of plasma and cerebrospinal fluid (CSF) samples obtained at sacrification following 2 months of phloridzin treatment. Results of (**a**) plasma total glucagon-like peptide-1 (GLP-1); (**b**) CSF total GLP-1; (**c**) plasma active GLP-1; (**d**) plasma total gastrointestinal inhibitory polypeptide (GIP); (**e**) CSF total GIP; (**f**) plasma active GIP; (**g**) plasma glucose; (**h**) CSF glucose. Data presented as mean ± SD with individual animal data and analysed by Kruskal–Wallis 1-way ANOVA with Wilcoxon signed rank tests for between-group comparisons. CTR—control group; STZ—streptozotocin-treated group; CTR+SGLTI—control group treated with phloridzin; STZ+SGLTI—streptozotocin-treated group treated with phloridzin.

**Figure 6 biomedicines-11-01025-f006:**
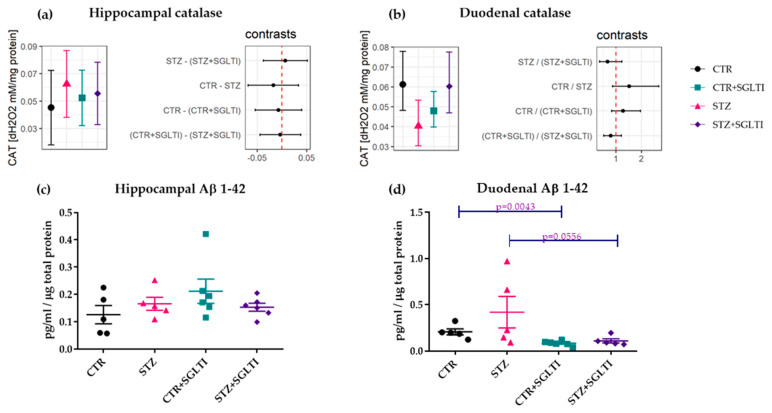
Analysis of hippocampal and duodenal soluble amyloid β (Aβ) 1-42 peptide content and catalase activity following 2 months of phloridzin treatment. The results of (**a**) hippocampal and (**b**) duodenal catalase activity are presented as least square mean estimates with 95% confidence interval (**left**) and estimates of contrasts (differences) with 95% confidence interval (**right**). The dependent variable (catalase activity) was adjusted for protein concentration (loading control) and baseline H_2_O_2_. The results of (**c**) hippocampal and (**d**) duodenal Aβ 1-42 level are presented as mean ± SD with individual animal data and analysed by Kruskal–Wallis 1-way ANOVA with Wilcoxon signed rank tests for between-group comparisons. CTR—control group; STZ—streptozotocin-treated group; CTR+SGLTI—control group treated with phloridzin; STZ+SGLTI—streptozotocin-treated group treated with phloridzin.

## Data Availability

Not applicable.

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
