# Peer review of "The Effect of the Sodium—Glucose Cotransporter Inhibitor on Cognition and Metabolic Parameters in a Rat Model of Sporadic Alzheimer’s Disease"

_biomedicines, 2023, doi:10.3390/biomedicines11041025_

Round 1
Reviewer 1 Report
Thank you for your hard work. The reviewer can definitely see a lot of work involved in this paper. I only have some minor comments:
1. would the gender or age of rats have an impact on this study?
2. would you study the optimized dosage and frequency for STZ and SGLTI treatment in future work?
Reviewer 2 Report
The manuscript by Alfani Abrar et al., titled “The effect of the sodium-glucose cotransporter inhibitor on cognition and metabolic parameters in a rat model of sporadic Alzheimer’s disease”, is an interesting study which is similar to previously published work. However, manuscripts need to be further strengthened.
Major comments
1. Explain the rationale for using 10mg phlorizin without performing a Dose–response test.
2. Authors should consider performing multiple concentrations of the drug.
3. Authors need to demonstrate the disease pathology in the presence or absence of drugs using histology or immunostaining in rat brain sections.
4. Authors should use the positive control similar to the previous studies.
5. Most of the error bars in the figures are overlapping, hence consider presenting data in Mean ± S.E.M
6. Authors need to measure HbA1c levels in STZ-infused rats.
7. Authors should verify the drug's effect on the sex of animals.
Minor comments:
1. Please give a reference / rational for deciding the dose of SGLT1 (10mg/kg).
Reviewer 3 Report
Major comment:
1. The authors have showed that long term treatment with SGLTi has no effect on cognition. Could the authors also check for the level of beta-amyloid and p-Tau in the brain? Although the authors have mentioned it in the limitation of the study, this is a relatively easy experiment to do.
Minor comment:
1. p values for all of the comparisons need to be shown in the figures 3g, and figure 5.
Author Response
"Please see the attachment."

Round 2
Reviewer 2 Report
I appreciate Alfani Abrar et al., for answering few of the comments. However, I still believe major question need to be answered with new data for previously suggested comments
Major comments
1. Authors should consider performing multiple concentrations of the drug.
2. Authors need to demonstrate the disease pathology in the presence or absence of drugs using histology or immunostaining in rat brain sections.
3. Authors should use the positive control similar to the previous studies
Reviewer 3 Report
The authors have added an experiment in doudenum, and it elevates the quality of the work. The manuscript can be accepted in the current form.
Round 3
Reviewer 2 Report
I still believe major question need to be answered with experiments for the following
1. Authors should consider performing multiple concentrations of the drug.
2. Authors need to demonstrate the disease pathology in the presence or absence of drugs using histology or immunostaining in rat brain sections.
3. Authors should use the positive control similar to the previous studies
Author Response
"Please see the attachment."
